# The Role of Gut Microbiota Biomodulators on Mucosal Immunity and Intestinal Inflammation

**DOI:** 10.3390/cells9051234

**Published:** 2020-05-16

**Authors:** Chiara Amoroso, Federica Perillo, Francesco Strati, Massimo Fantini, Flavio Caprioli, Federica Facciotti

**Affiliations:** 1Department of Experimental Oncology, IEO European Institute of Oncology IRCCS, 20139 Milan, Italy; chiara.amoroso@ieo.it (C.A.); federica.perillo@ieo.it (F.P.); Francesco.strati@ieo.it (F.S.); 2Gastroenterology Unit, Duilio Casula Hospital, AOU Cagliari, 09042 Cagliari, Italy; massimoc.fantini@unica.it; 3Department of Medical Science and Public Health, University of Cagliari, 09124 Cagliari, Italy; 4Department of Pathophysiology and Transplantation, Università degli Studi di Milano, 20135 Milan, Italy; flavio.caprioli@unimi.it; 5Gastroenterology and Endoscopy Unit, Fondazione IRCCS Cà Granda, Ospedale Maggiore Policlinico, 20135 Milan, Italy

**Keywords:** inflammatory bowel diseases, gut microbiome, live biotherapeutic products, FMT

## Abstract

Alterations of the gut microbiota may cause dysregulated mucosal immune responses leading to the onset of inflammatory bowel diseases (IBD) in genetically susceptible hosts. Restoring immune homeostasis through the normalization of the gut microbiota is now considered a valuable therapeutic approach to treat IBD patients. The customization of microbe-targeted therapies, including antibiotics, prebiotics, live biotherapeutics and faecal microbiota transplantation, is therefore considered to support current therapies in IBD management. In this review, we will discuss recent advancements in the understanding of host−microbe interactions in IBD and the basis to promote homeostatic immune responses through microbe-targeted therapies. By considering gut microbiota dysbiosis as a key feature for the establishment of chronic inflammatory events, in the near future it will be suitable to design new cost-effective, physiologic, and patient-oriented therapeutic strategies for the treatment of IBD that can be applied in a personalized manner.

## 1. Background

The pathogenesis of inflammatory bowel diseases (IBD) is still incompletely understood. However, the combination of genetic and environmental factors in the context of host–microbe interactions appears to trigger IBD-initiating events. Impairment of intestinal barrier functions, which results in the translocation of gut microbes, promotes the hyper-activation of the mucosal immune system and the production of pro-inflammatory cytokines that altogether contribute to fuelling the inflammation observed in IBD patients [1,2,3]. The gut microbiome is a complex microbial ecosystem that co-evolved a mutualistic relationship with the host complementing its functions through dietary fibre fermentation, pathogen defence and biosynthesis of vitamins and essential metabolites. The mutual interaction between the gut microbiota and the host is further highlighted by its role in sustaining the maturation and functioning of the host’s immune system contributing to host’s homeostasis [4]. Indeed, the intestine and its associated immunological components have to deal with several, in some cases dichotomous, tasks. Apart from all the functions related to digestion and absorption of nutrients, the intestine has to be tolerant towards mutualistic/commensal microorganisms and to keep control over pathobionts (i.e., those resident microbes with pathogenic potential), preventing microbial overgrowth and invasion of the epithelial intestinal barrier. In turn, the gut microbiota has to modulate and regulate several aspects of host’s immune system towards tolerance rather than responsiveness. Any disruption of this delicate equilibrium has potentially pathological consequences on the health status of the host. Dysbiosis (i.e., altered microbial composition) of the gut microbiota can lead to chronic inflammation as observed in IBDs. IBDs appear to be caused by a dysregulated immune response to commensal microorganisms harbouring virulence traits in their genome (the so-called pathobionts) in genetically susceptible hosts. In this review, we will discuss the relationship between the microbiota and the immune system in IBD patients. We will recapitulate the mechanisms by which the gut microbiota and the immune system reciprocally influence their functions in homeostasis and during IBD as well as the basis for therapeutic restoration of homeostatic immune function by manipulating the gut microbiota through existing microbe-targeted therapies, including antibiotics, prebiotics, probiotics, and faecal microbiota transplantation. 

## 2. Mucosal Immune Dysfunctions and Dysbiotic Microbiota in IBD

It is now clear that IBD is a polymicrobial disease with a combination of various gut microbial factors, abnormal immune responses and a weakened intestinal mucosal barrier leading to aberrant host immune responses against commensal bacteria [5]. Over the last years, through genome-wide association studies (GWAS) in diverse populations, genetic variants and candidate gene networks that affect host-microbe interactions such as Toll-like receptor (TLR) and nucleotide-binding oligomerization domain (NOD)-like receptor (NLR) signalling have been identified [6]. IBD loci are also markedly enriched in genes involved in primary immunodeficiencies, which are characterized by a dysfunctional immune system. These genes correlate with reduced levels of circulating T-cells (ADA, CD40, TAP1/2, NBS1, BLM, DNMT3B), or of specific subsets such as T helper (Th)-17 (STAT3), memory (SP110), or regulatory T-cells (STAT5B) [7]. Intestinal epithelial barrier integrity defects are frequently observed in IBD. Indeed, the epithelial barrier plays a critical role in maintaining intestinal homeostasis because it lies at the interface between luminal microbes and the host immune system, while also being the first site of exposure to many of the environmental factors that can act as triggers of disease activity [8]. Accordingly, several genes closely related to IEC-specific functions, such as *ATG16L1*, *KCNN4*, *XBP1*, may promote the susceptibility to IBD [9]. 

It has been observed that the TNF-α-TNFR2 signalling pathway in intestinal epithelial cells (IECs) increases the expression of myosin light chain kinase (MLCK), and thereby disrupt the assembly of the tight junctions (TJ) [10], suggesting that proinflammatory cytokine may further enhance the “leakiness” of the epithelial layer. Moreover, inflammation may induce the so called “depletion of goblet cells” determining the loss of proper mucin secretion [11] and a deregulated IL-7 cytokine production, leading to chronic inflammation [12]. Expression of antimicrobial peptides is mostly reduced in IBD. IBD candidate risk loci, in particular those with mutations in the *CARD15* gene, encoding the bacterial-sensing protein NOD2, have been associated with reduced α-defensin production (i.e., DEFA5 and DEFA6) in both adult and pediatric ileal patients with CD [13,14].Also the CD risk allele *ATG16L1* predisposes Paneth cells to lose their ability to form normal intracellular granules [15], resulting in a lower production of antimicrobial peptides [16]. Thus, defects in epithelial cell barrier function lead to chronic exposure to bacterially derived molecules leading to the destructive intestinal inflammation that characterize IBD [17].

Th17 cells abundantly exist in the LP of the small intestine [18] and they not only protect the host against infection, but their hyper-activation also cause autoimmune inflammation in the gut [19]. The gut microbiota has a very strong influence on the frequency of Th17. Segmented filamentous bacteria (*SFB*) presence, for example, is sufficient to induce the appearance of classical mucosal CD4+ Th17 cells coproducing interleukin (IL)-17 and IL-22 [20] with distinct T cell receptor (TCR) clonotype [21]. A functional plasticity of Th17 cells towards a Th1 lineage has been demonstrated, which is dependent on the presence of IL-12 and IL-23 produced by antigen-presenting cells (APCs) in response to bacteria-derived signals, both in murine models and IBD patients [22]. Nizzoli and colleagues recently demonstrated that pathogenicity of IL-17-secreting cells isolated from Crohn’s disease (CD) patients and from colitic animals were directly dependent on Interferon-γ (IFN-γ) secretion, as demonstrated by the reduced colitogenic activity of Th17 cells isolated from IFN-γ-/- mice [22]. Paradoxically, secukinumab, a monoclonal antibody used as treatment for dermatological and rheumatological disease which acts by blocking of the IL-17 pathway, is associated to IBD onset in approximately 1% of patients [23,24]. This contradictory effect may be due to the fact that IL-17 seems to act as protector against inflammation, contributing to the inhibition of the Th1 response and maintaining the integrity of the enterocyte’s epithelial barrier and intestinal homeostasis [25]. 

Alterations in the composition of the commensal microbiota, such as bifidobacteria and lactobacilli, can influence also the frequency of mucosal Treg cells, which are known to play a pivotal role in the pathogenesis of IBD [26], as demonstrated by the fact that mice lacking Treg cells develop spontaneous colitis [27]. Similarly, children with mutations in the IL-10 receptor develop early-onset CD [28]. Nevertheless, increased numbers of Foxp3+Tregs and high levels of their signature cytokines TGF-β and IL-10 have been reported in inflamed intestinal lesions of IBD patients as well as in colitic animals [29] suggesting that inflammation can promote Treg expansion and accumulation in inflamed lesions. Gram-positive commensal bacteria play a dominant role in maintaining Treg homeostasis [30], as confirmed by the experiments where reconstitution of Germ-Free (GF) mice with Gram-positive spore-forming microorganisms, restored the Treg population [31]. It has been demonstrated that the culture supernatant of IECs from *Clostridium*-colonized mice markedly enhanced the differentiation of Foxp3-expressing cells, showing that *Clostridia* activate IECs to produce TGF-β and other Treg-inducing molecules within the colon [30]. In particular, *F. prausnitzii* skewed human dendritic cells (DCs) to prime IL-10-secreting T cells and to express a unique array of potent type 1 regulatory T (Tr1)/Treg polarizing molecules such as IL-10, IL-27, CD39, indoleamine 23-dioxygenase 1 (IDO-1) and programmed death-ligand 1 (PDL-1). Following TLR4 stimulation, *F. prausnitzii* is also able to reduce the up-regulation of co-stimulatory molecules as well as the production of the pro-inflammatory cytokines IL-12 (p35 and p40) and Tumor Necrosis Factor (TNF)-α [32]. These data suggest that the composition of the gut microbiota may affect human colonic homeostasis by acting on the DCs- Treg cells induction axis [32]. 

Inducible (i)Treg cells, suppressive cells which develop from mature CD4+ conventional T cells outside of the thymus and which are involved in mucosal tolerance, are induced and maintained by gut microbes [33,34]; these cells have been thoroughly studied in the pathogenesis of IBD. The decreased percentage of iTreg may lead to autoimmune responses and tissue damage in the acute phase of IBD, although it has not been conclusively ruled out if iTreg takes part in promoting intestinal homeostasis during the recovery stage [35].

Invariant natural killer T (iNKT) cells are critical players in the mucosal immune responses [36], but their role in IBD has not been completely elucidated. iNKT cells have been reported to contribute to experimental intestinal inflammation [37], and those isolated from IBD patients have a pro-inflammatory phenotype manifesting pathogenic features upon exposure to intestinal mucosa-associated microbiota [38]. However, it has also been shown that iNKT cells contribute to intestinal homeostasis by interacting with CD1d-expressing, IL10 producing, epithelial cells [39] and that iNKT cells protect mice from experimental colitis [40,41,42], albeit in IBD patients a protective role for iNKT cells has not been proven yet. *B. fragilis* effectively regulates iNKT cell proliferation during neonatal development, thanks to the inhibitory effects of its glycosphingolipid GSL-Bf717. When *B. fragilis* is present in the eubiotic microbiota, total colonic iNKT cell numbers are restricted into adulthood by recognition of GSL-Bf717, and the host is protected against experimental oxazolone-induced colitis [36]. Moreover, *B. fragilis* colonization can reverse CD4+T-cell defects and Th1/Th2 imbalance in GF mice [43] and can protect from experimental colitis induced by *Helicobacter hepaticus*. 

Innate lymphoid cells (ILCs), in particular ILC3s, are also thought to be important in the pathogenesis of IBD [44]. IL-17-producing ILC3 cells are increased in inflamed intestines in patients with Crohn Disease (CD) [44]. IL-22 +ILC3s deficiency, found in both the intestinal mucosa of animal models and patients with IBD, causes intestinal mucosal barrier damage, leading to exposure of intestinal tissue to a large number of antigens [45]. Moreover, production of IL-22 by ILC3s is required for protective immunity towards pathobionts, such as *Citrobacter rodentium*, since mice lacking ILC3s or IL-22 quickly succumb to the infection [46]. Macrophages and DCs are also involved in gut bacteria recognition during intestinal inflammation [29,47]. In IBD patients, during the inflammatory processes, monocytes move into the intestine to differentiate into macrophages and DCs and the latter express higher levels of TLR-2 and TLR-4, which may contribute to an altered immune response to commensals, and of CD40 [48]. Consequently, an increased production of proinflammatory cytokines, such as IL-1, IL-6, TNF, IL-18 and members of the IL-12 family is observed [48]. Furthermore, neutrophils show an important antimicrobial function that relies on the formation of neutrophil extracellular traps (NETs) [49]. The generation and/or clearance of aberrant NETs has been associated with several immune diseases [50] including Ulcerative Colitis (UC), where excessive NETs formation in response to TNF-α stimulation can amplify pathogenic signals in the gut [51]. It has been shown that *Candida albicans* might interact with mucosal innate immune cells through the pathways associated with Dectin-1 in macrophages [52] and TLR4 in neutrophils and by inducing the proliferation and differentiation of B-lymphocytes accompanied with increased number of Immunoglobulin (Ig)A-secreting plasma cells [53]. Eukaryotic viruses are also involved in intestinal inflammatory processes. Infection with the murine norovirus in genetically predisposed mice triggers the alteration of Paneth cells activity and the inflammatory response when treated with dextran sodium sulphate (DSS), by modulating the cytokines TNF-α and IFN-γ, as well as by inducing alterations in the composition of the commensal microbiota [54]. 

## 3. Manipulation of the Gut Microbiota for Therapeutic Purposes in Intestinal Inflammation

The conventional treatments for IBD mainly aim at suppressing the enhanced immune response by the use of steroids, thiopurines, biologic medicines (i.e., anti-TNF, or anti-IL-12/23), small molecules including anti-Janus kinases (JAK) inhibitors, and molecules blocking the homing of pathogenic immune cells in the inflamed gut (i.e., anti-integrins) [55]. The use of anti-TNF agents has dramatically changed the management of IBD in the last decades [56] but did not change the long-term course in certain subsets of paediatric CD patients [57] as well as the frequency of hospitalization in CD patients [58]. Furthermore, the availability of multiple compounds for IBD treatment advocates the identification of variables or bio-markers able to predict therapeutic outcomes to find the best candidate drug to a given treatment. A phase-IV explorative prospective interventional trial aimed at identifying immunological variables associated with response to vedolizumab (VDZ) in patients with UC and CD demonstrated that the baseline immunological profiling of circulating and mucosal Th lymphocytes, in particular CXCR3+ Th cells, are associated with both clinical and endoscopic response to VDZ [59]. In addition, the baseline serum levels of IL-6 and IL-8 were associated with positive clinical outcomes in UC patients treated with VDZ; thus, an early assessment of these cytokines in UC patients treated with VDZ could lead to significant savings in terms of health care resources [60]. Accordingly, a recent work demonstrated that the levels of IL-6 may allow for the prediction of clinical response at 12 months of biological therapy thus helping to design personalized treatment strategies [61]. These biological drugs may also act by restoring the composition of the gut microbiota, as we discuss below, such as adalimumab, which is able to control inflammation by normalizing both the levels of C reactive protein (CRP), and the intestinal microbial community structure [62]. 

However, these immunosuppressive therapies are not always effective, are quite expensive and potentially induce serious side effects. Therefore, it is desirable to develop personalized strategies to identify which patients should be treated with which drugs [63] by leveraging a more physiological approach, patient centred, with limited toxicity and high cost-effectiveness. Based on the current knowledge in the pathogenesis of IBD, microbe-targeted therapies aimed at restoring gut dysbiosis and immune homeostasis in IBD patients seems to be promising therapeutic options. Indeed, several therapies among which antibiotic treatments, probiotics and prebiotics administration as well as FMT have been explored to target and modulate gut microbiota composition, including both microbial physiology and/or their metabolites, that cause or contribute to inflammation directly or indirectly [64] (Figure 1). 

### 3.1. Antibiotics

Since it is now acknowledged which bacterial species may have a role in IBD patients, the selective administration of different antibiotics has been used to control intestinal inflammation. Accordingly, ciprofloxacin, metronidazole or rifaximin are used to reduce the abundance of pathobionts [65] (Table 1). Treatment of colitic mice with minocycline, a semi-synthetic second-generation tetracycline, induced a reduction of pro-inflammatory cytokines [66]. Ciprofloxacin and metronidazole are commonly administered to CD patients [67] and are effective for anal lesions and delay of postoperative recurrence in CD [68] by reducing TNF-α, IL-1β and IL-8 [69] or by inducing long-term changes in the immune phenotype of Tregs and naive T-cells [70]. Rifaximin, a non-absorbable antibiotic, showed an excellent safety profile and coupled with a reduction of colonic inflammation and bacterial translocation in the mesenteric lymph nodes (MLNs) [71]; however, it does not yet have validated efficacy [72]. Despite some favourable clinical effects, the use of broad-spectrum antibiotics strongly affects the composition of the gut microbiota hampering the reconstitution of the gut microbiota and promoting a pro-inflammatory phenotype in the long term. Indeed, we recently demonstrated how a short-term treatment with broad-spectrum antibiotics profoundly affected the frequency and function of intestinal iNKT cells, but not of CD4+ T cells in the absence of intestinal inflammation [37]. Reconstitution of the gut microbiota after antibiotic treatment was sufficient to imprint colonic iNKT and CD4+ T cells toward a Th1-Th17 pro-inflammatory phenotype, capable to aggravate clinical conditions upon intestinal inflammation [37].

### 3.2. Probiotics and LBPs

Probiotics are living microorganisms that can be used as non-pharmacological methods to promote gut health and potentially modulate dysbiosis in IBD [112]. Their mechanism of action is largely strain-dependent, although their beneficial effects mostly depend on their metabolisms and metabolic by-products (e.g., SCFAs, bacteriocins, hydroperoxides, secondary bile acids, and lactic acids) capable to promote the release of cellular components in the gut environment which in turn can activate immune responses [113]. Probiotics may be sampled by M cells in the Peyer’s patches [114] and modulate the activity of APC [115]. The interaction between probiotics and DCs may influence the subsequent antigen-specific T cell response towards Th1, Th2, Th17 or Treg cells, although some strains of lactobacilli *(L. plantarum*) regulates Treg frequency and DCs activation even in the absence of M-dependent sampling [73]. Different strains of *Lactobacillus* and *Bifidobacterium* showed significant capability to reduce pro-inflammatory IL-6 and IL-17 levels [74], to restore the Treg/Th17 balance [75] via inhibition of the NF-κB signalling pathway in macrophages and epithelial cells and to control the overgrowth of pathobionts belonging to *Enterobacteriaceae* [76,77]. *E. coli Nissle 1917* (*EcN),* can colonize the intestine and perform several documented protective functions being successfully used to extend remission phases in IBD patients in clinical routine [116]. *EcN* flagellin is able to induce a strong activation of TLR-5 and therefore to enhance IL-22 production, a cytokine mediating epithelial reconstitution [78] and promoting intestinal immune homeostasis via Treg expansion [79]. In addition, probiotics can indirectly affect pathobionts colonization via host’s PPAR-γ signalling [117]. VSL#3 is a high-concentration probiotic preparation of eight live freeze-dried bacterial probiotic species including lactobacilli (*L. casei, L. plantarum, L. acidophilus*, and *L. delbrueckii* subsp. *bulgaricus*), bifidobacteria (*B. longum, B. breve*, and *B. infantis*), and *Streptococcus salivarius* subsp. [118], capable to regulate microbiota composition [119] by reducing luminal oxygen and inhibiting aerobic *Enterobacteriaceae* and immune responses [80]. The VSL#3 treatment determines anti-inflammatory effects in experimental colitis, as evidenced by reduced disease activity index (DAI) score, histological activity index (HAI) score and myeloperoxidase (MPO) activity [120]. Furthermore, in different randomized, double-blind, placebo-controlled trials the use of VSL#3 in UC patients showed significant effects in terms of clinical remission and clinical response during active UC with no side-effects [81,121]. However, the efficacy of probiotics in IBD remains uncertain, since different meta-analyses showed that there were no significant differences of remission, relapse, and complication rate between probiotics and placebo group, thus, more evidences from randomized clinical trials (RCTs) are required [122,123].

Not only bacteria but also fungi are used as probiotics. *Saccharomyces boulardii* is a well characterized probiotic yeast often used to alleviate gastrointestinal (GI) tract disorders [82]. Nevertheless, probiotics effects are often transient and limited in most IBD subsets because of i) inability to replace/restore the microbial species depleted in IBD patients [124], ii) colonization resistance, since individual immunological status and mucosal microbial features are associated with probiotics persistence [125], iii) treatment timing and proper delivery mode [126]. Therefore, a new era in probiotic research is addressing specific patient’s needs towards the comprehension of a wider microorganism’s range with potential health benefits. These organisms are known as live biotherapeutic products (LBPs). LBPs conform to the normal definition of probiotics but they differ from them because they also include genetically modified microorganisms (GMMs) and because their use is addressed under a pharmacological point-of-view [126]. Despite remaining safety and environmental concerns, GMM can be used as vehicle for delivering a range of anti-inflammatory molecules such as anti-TNF, IL-10-, IL-27-, elafin-secreting *Lactococcus* species [127] and efficiently alleviate mucosal inflammation by promoting a homeostatic immunologic profile.

### 3.3. Prebiotic Diets and Synbiotics

Prebiotics are non-digestible dietary compounds that stimulate the growth and activity of probiotics, conferring a competitive advantage to commensal bacteria capable of metabolizing these substrates or by increasing the production of beneficial metabolic products that result from their fermentation [3]. Since prebiotics are resistant to hydrolysis by human alimentary tract enzymes, they can be fermented by colonic anaerobic bacteria to produce metabolites important for host physiology such as SCFA [128,129]. A high-fiber diet has been reported to protect against the development of experimental intestinal inflammation [130] and to directly interact with host cells modulating immune responses [131]. High doses of pectin decreases the expression of colonic pro-inflammatory mediators, including IL-1β and IL-6, [83]. Oral supplementation with 2-fucosyllactose significantly decreased the severity of colitis in IL-10^−/−^ mice, reducing the expression of IL-1β and IL-6 and increasing TGF-β expression and expansion of the propionate-producing commensal *Ruminococcus gnavus* [84]. *Dictyophora indusiate* polysaccharides (DIP) demonstrated to have therapeutic potential in the restoration of antibiotic-induced gut dysbiosis and inflammatory responses in BALB/c mice [85] and in DSS-induced colitis mice [86]. Polysaccharide, alcoholic and whole extracts of the edible mushroom *Hericium erinaceus* have been reported to reduce the abundance of pathobionts and to promote the enrichment of some health-promoting bacteria, (i.e., *Bifidobacterium, Parabacteroides, Lactobacillus, Coprococcus)* [87]. Furthermore, these extracts reduced the levels of serum IL1-α, IL-2, IL-8, IL-10, IL-11, IL-12, TNF-γ, TNF-α, vascular endothelial growth factor (VGEF), macrophage inflammatory protein-α (MIP-α), macrophage colony-stimulating factor (M-CSF) and MPO as well as the proportion of TNF-α and NF-κB p65- positive cells and Tregs increased [88]. Also grape polyphenols showed prebiotic activities, promoting the growth of *Bifidobacerium* and *Lactobacillus* while inhibiting that of *Clostridium histolyticum* [132]. Muscadine grapes or wine phytochemicals are capable to decrease MPO activity as well as colonic levels of IL-1β, IL-6, and TNF-α in experimental DSS-colitis [89]. Whole muscadine grapes (FMG) or dealcoholized muscadine wine (DMW) supplementation in colitic animals decreases the relative abundance of *Clostridium* and *Akkermansia*, increases abundances of *Roseburia, Anaerotruncus*, and *Coprococcus,* resulting in higher faecal levels of butyric acid, acetic acid and fecal IgA, suggesting a more robust activation of the adaptive immune system [90]. The combination of both prebiotics and probiotics, called “synbiotic”, can provide beneficial effects to the host and improve the viability of its constituents [133]. Given that IBD patients harbour less health-promoting bacteria, administration of synbiotics may improve treatment with probiotics or LBPs. *B. coagulans* MTCC585, in combination with the prebiotic whole plant sugar cane fibre (PSCF), has synergistic effects in colitic mice being sufficient to modulate serum IL-1β, IL-10, and CRP levels and raising the SCFA profile [91]. Similarly, the synbiotic *B. coagulans* combination with the green banana resistant starch (GBRS) demonstrated to prevent intestinal inflammation through the suppression of IL-1β and to increase IL-10 levels [92]. A recent randomized placebo-controlled study evaluated the efficacy of a synbiotic therapy in mild-to-moderately active UC by administrating six probiotic strains with the prebiotic fructo-oligosaccharide (FOS). The study showed an amelioration of the disease status of the synbiotic treated UC patients and a significant reduction of systemic inflammation as measured by serum CRP levels. These results support the use of synbiotics as an effective strategy to prevent the exacerbation of the disease in patients with mild-to-moderately active UC [93]. Nevertheless, although prebiotics have an excellent safety profile, they have been associated with symptoms of dose-dependent abdominal pain, flatulence, bloating, and diarrhea [134]; therefore, insoluble fibre intake is discouraged in the case of significant intestinal stenosis in IBD patients [135].

### 3.4. Postbiotics

Bacterial viability is not always required to promote health-promoting effects, leading to the hypothesis that bioactive compounds derived from probiotics fermentation processes, the so-called postbiotics, might suffice to promote a healthy status [136] and be a safer alternative for patient treatments. Postbiotics comprise SCFAs, enzymes, peptides, teichoic acids, peptidoglycan-derived muropeptides, endo- and exo-polysaccharides, cell surface proteins, vitamins, plasmalogens, and organic acids [137] and, in the majority of cases, are derived from *Lactobacillus* and *Bifidobacterium* strains, but also from *Streptococcus* and *Faecalibacterium* species [94]. The mechanisms of action of postbiotics are not well characterized, but it has been hypothesised that they may influence the host cellular pathways involving proliferation, differentiation, migration and cellular death as well as the maturation/function of mucosal and systemic immunity [138]. The use of viable biotherapeutic products might cause problems related to safety for human use. Indeed, the use of *L. plantarum NCIMB8826*, *L. rhamnosus GG* (LGG) and *L. paracasei B21060* showed deleterious side-effects on inflamed IBD-derived tissues [139,140]. Incubation of IBD-derived organ cultures with *L. paracasei* supernatants resulted instead in a significant reduction in TNF-α production as well as in most of the cytokines and chemokines involved in the pathology of IBD, including chemokine (C-C motif) ligands (CCL)-4, CCL2, IFN-γ and IL23p40 [140]. Similarly, the spent medium of *L. rhamnosus GG* (LGG) promoted IgA secretion decreasing the susceptibility to intestinal injury and colitis in adult mice [95], a mechanism mediated by the *L. rhamnosus GG* protein p40 [96,97,98]. Postbiotics from *L. plantarum RG14* also modulated leukocyte, lymphocyte, basophil, neutrophil and platelets counts, increased IL-6 mRNA and decreased of IL-1β, IL-10, TNF mRNA levels [99]. Conjugated linolenic acids (CLNA), a positional and geometric isomer of unsaturated fatty acids, obtained from the fermentation of *Lactobacillus plantarum ZS2058* are also potential postbiotic agents. The isomers of c9, t11, c15-CLNA (CLNA-1) and t9, t11, c15-CLNA (CLNA-2) significantly attenuated the level of MPO and of proinflammatory cytokines (TNF-α, IL-1β, and IL-6) while upregulated the expression of IL-10 and the nuclear receptor PPARγ [100]. Oral administration of secondary bile acid ursodeoxycholic acid (UDCA) and its taurine (TUDCA) or glycine conjugates (GUDCA) to colitic mice were also shown to control intestinal dysbiosis and the expression of inflammatory chemokines and cytokines, such as chemokine (C-X-C motif) ligand 1 (CXCL1), granulocyte colony-stimulating factor (G-CSF), and IL-6. At systemic level, lower levels of CXCL1 and G-CSF, but not IL-6, were detected in the serum of bile acid-treated mice than in that of the placebo-treated group [101]. Peptides derived from *Bacteroides fragilis YCH46* (peptide B12) [141] and from probiotic *Bifidobacterium longum* subsp. *longum ATCC15707* (peptide B7) [142,143,144] can influence the immunoregulatory capacity of human APCs (including B-cells, monocytes, plasmacytoid DCs (pDCs), and conventional DCs (cDCs). Both peptides are capable to increase the pro-inflammatory profile of IBD-derived APC, as demonstrated by the enhanced expression of human leukocyte antigen-DR (HLA-DR) on cDC prompted by peptide B7, which also expanded IL-1β production of B-cells and on pDC by B12. This phenomenon was not observed in APC derived from healthy individuals, suggesting that differential immune mechanisms between healthy controls and IBD patients may abrogate the immunomodulatory tolerogenic potential of bioactive peptides from the gut microbiota in IBD [145]. In experimental colitis models, butyrate, a SCFA, produced by *F. prausnitzii* maintained Th17/Treg balance and exerted significant anti-inflammatory effects via inhibition of the IL-6/signal transducer, the STAT3/IL-17 pathway and promoted Foxp3 expression by targeting histone deacetylase 1 (HDAC1) [102]. Butyrate-derivates are currently studied for oral administration to patients such as the N-(1-carbamoyl-2-phenylethyl) butyramide (FBA) which showed similar health-promoting activities compared to butyrate [103]. Xylan butyrate ester (XylB) is a butyrate-releasing polysaccharide derivative. XylB treatments reversed the imbalance between pro- (IL-1β, TNF-α, and IL-17A) and anti-inflammatory (IL-10) cytokines in experimental colitis. Moreover, XylB rebalanced the gut microbiota and significantly decreased the relative abundance of the genera *Oscillibacter*, *Ruminococcaceae*, *Erysipelatoclostridium*, and *Defluviitaleaceae* [104]. The use of postbiotics has been tested also in a prospective-randomised-placebo-controlled study with IBD patients. The effect of a microencapsulated form of sodium butyrate (MSB) was evaluated. MSB induced significant changes in the gut microbiota of IBD patients by increasing the abundance of SFCA-producing bacteria. This study showed that MSB supplementation produced a mimicking prebiotic effect increasing the production of endogenous and physiological SCFAs with a marked improvement of quality of life and reduction of inflammatory markers [105].

### 3.5. Faecal Microbiota Transplantation

Gut microbiota manipulation through faecal microbiota transplantation (FMT), i.e., the procedure of transferring the microbial ecology of a healthy donor into a patient to treat microbial dysbiosis [146], is gaining great attention by the medical community. Currently, FMT has proved to be highly successful in treating recurrent and antibiotic refractory *Clostridiodes difficile* (*C. difficile*) infection (CDI), with cure rates approaching 90% [147]. Nevertheless, while it is clear that in CDI FMT efficiently eliminates the pathogen and its virulence factors, [148], less is known on the mechanisms behind the therapeutic effects of FMT in IBD. We showed that therapeutic FMT administered during acute experimental colitis could directly modulate both innate and adaptive mucosal immune responses towards the control of intestinal inflammation. Therapeutic FMT not only is able to reduce colonic inflammation, as demonstrated by decreased levels of the pro-inflammatory cytokines TNF, IL1-β and IFN-γ, but also to initiate the restoration of intestinal homeostasis through the simultaneous activation of different immune-mediated pathways. Indeed, higher amounts of colonic IL-10 as well as increased frequencies of IL-10-producing APC and CD4+ T and iNKT cells were observed in FMT-treated mice as compared to DSS-treated mice. Furthermore, FMT treatments reduced the ability of DCs, monocytes and macrophages to present major histocompatibility complex (MHC-II) dependent bacterial antigens to colonic T cells. The beneficial effects of FMT during intestinal inflammation were also associated with a reshuffling of the intestinal microbial communities towards restoration of functional normobiosis [106]. Furthermore, we also showed that therapeutic FMT administration in the context of a chronic experimental colitis setting, a condition more similar to that of IBD patients, stably decreased colonic inflammation by modulating the expression of pro-inflammatory genes, such as Ifng, Tnf, Il1b, Il-17, and Il-6 [107]. Mice receiving FMT displayed a lower proportion of CD4+ T and CD8+ T cells expressing the cytotoxicity-associated molecule CD107a as well as reduction of colonic MHC-II-expressing professional APCs [107]. Recently, it was also examined the impact of transferring the gut microbiota from healthy or IBD donors into GF mice to determine the homeostatic intestinal T cell response to each microbiota. The transfer of IBD microbiota into GF mice increased the numbers of intestinal Th17 cells and Th2 cells and decreased numbers of RORγt+ Treg cells. These data further support the concept that only an eubiotic core ecology is able to induce or maintain an anti-inflammatory polarized mucosal immune response [149]. Most of the available data regarding FMT in IBD come from studies in UC patients [150]. In a single-centre, prospective, open-label pilot study was tested the impact of preparation and donor characteristics on FMT success. Immune cell profiling was performed on mucosal biopsies before and after FMT to assess its impact on mucosal T cell immunity. Analysis of CD4+ T cell cytokine production revealed a significant reduction in IFN-γ in Tregs at week 4 compared to time of transplant, however no difference in IL-4, IL-17, IL-22 or Th17 was reported [108]. Moreover, specific members of the gut mycobiota can play a protective function in the gut [151]. Getting a better knowledge on host response to these organisms might harbour potential predictive markers on the outcome of microbiome-based therapies and should be further explored in future FMT trials [152].

Different RCTs have been performed so far in IBD patients, with those performed in mild-moderate UC giving the best results [153,154,155,156] while few data are available on CD patients [150]. All these RCTs demonstrated efficacy of FMT over placebo, although there were many inconsistencies in some studies where UC patients receiving FMT from healthy donors and autologous FMT did not show statistically significant difference in clinical remission [154,156]. 

Since almost all studies performed to date have assessed the role of FMT in remission induction for IBD, with a paucity of literature on the potential of FMT as a maintenance therapy, at present several clinical trials are evaluating the FMT influence on the immune system and clinical outcomes. The STOP-Colitis study [109] is evaluating not only the efficacy and safety of FMT in UC patients, but also the colonic immune profile of recipients before and after FMT. Furthermore, the Chinese FMTFUC study [110] will evaluate local and systemic inflammatory markers in UC FMT-treated patients. Finally, the University of Vermont Medical Center [NCT02390726] is assessing inflammatory markers pre- and post-FMT as well as changes in the host immune response via measurement of both mucosal and peripheral T-cells populations (Th1, Th2, Th17) using mucosal biopsies and blood samples respectively [157].

Unfortunately, at present, most of the available data regarding FMT in IBD is in UC patients and no RCTs results are available for CD patients, although the IMPACT-Crohn study [111] is assessing the impact of FMT on CD analysing also circulating and colonic lymphocyte levels in comparison with sham transplantation. 

## 4. Conclusions and Future Perspectives

IBD onset is caused by dysregulated immune responses to the gut microbiota in genetically susceptible hosts [158]. Although, the clinical heterogeneity of IBD patients as well as their unique gut microbiota features may reduce the efficacy of current treatments, selection of therapeutic targets based on individual’s microbiota patterns and immunological profiles will be important for the design of personalised and effective intervention strategies in the near future. To date, IBD patients are treated with standard anti-TNF agents and immunomodulators further supported by probiotics, prebiotics, antibiotics treatments or faecal microbial transplantation on an empiric base. The customization of these different treatments based on the clinical [59] and dysbiotic status of individual patients should be implemented in the current clinical practice to promote homeostatic immune responses in order to manage IBD in a more effective, physiologic, and patient-oriented manner.

## Figures and Tables

**Figure 1 cells-09-01234-f001:**
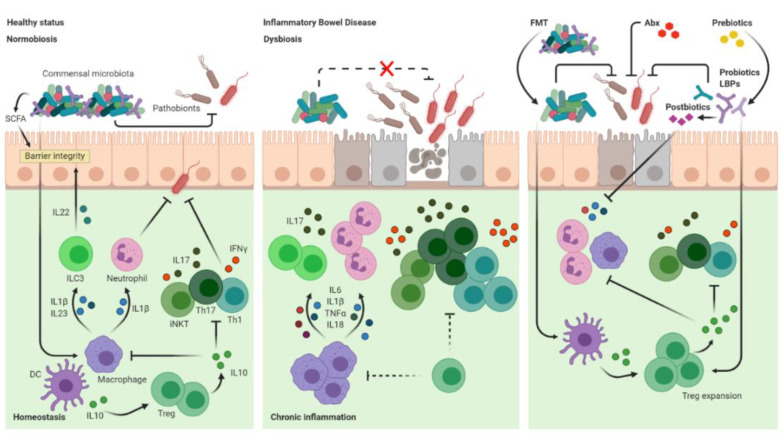
Homeostatic and pro-inflammatory role of the gut microbiota and microbe-targeted therapies for IBD management. During homeostasis (left panel), the recognition of specific PAMPs from the gut microbiota induces antigen presenting cells (APCs) like macrophages and dendritic cells (DC) to produce IL1β and IL23. Subsequently, different cell-types such as neutrophils, Th17 and ILC3 cells express cytokines (e.g., IL17 and IL22) that control the expansion of commensals and the potential invasion of microbes that could be harmful (pathobionts). The production of short-chain fatty acids (SCFAs), by the gut microbiota, in concert with IL22, enhances intestinal epithelial cell barrier function. Furthermore, commensals such as bifidobacteria, PSA+ *B. fragilis* and *Clostridium spp.* stimulates APCs to promote anti-inflammatory IL10 Treg responses regulating iNKT, Th1 and Th17 responses. In inflammatory bowel disease (IBD) (central panel) a combination of genetic and environmental factors lead to depletion of protective bacteria and the enrichment of colitogenic pathobionts, resulting in chronic inflammation due to hyper-activation of T helper 1 (Th1) and Th17 cells as well as aberrant innate, pro-inflammatory responses. Dashed lines show impaired responses. The use of microbe-targeted therapies (right panel) to restore homeostatic immune responses contribute to the expansion of anti-inflammatory responses by Treg cells and the modulation of pro-inflammatory cytokines release.

**Table 1 cells-09-01234-t001:** Microbe-targeted therapies for IBD.

Authors	Model/Study Case	Treatment	Outcomes	References
Antibiotics				
- N. Garrido-Mesa et al.	- TNBS-induced colitis Wistar rats and DSS-induced colitis C57BL/6J mice - Intestinal epithelial cells Caco-2	- Minocycline	- Reduced inflammatory infiltrate, colonic MPO activity, TNFα, IL-1β levels and colonic iNOS expression - Downregulated IL-8, IL-17, MCP-1, CINC-1, ICAM-1 - Reversed MUC-2 and TFF-3 reduction	[66]
- G. Lahat et al. - E. Becker et al.	- TNBS-induced colitis in BALB/c mice	- Ciprofloxacin - Metronidazole	- Reduced TNF-α, IL-1β and IL8 - Reduced recruitment of neutrophils - Reduced the levels of NFκB - Anti-inflammatory profile in Tregs after treatment; in naive T-cells observed only after the recovery period	[69,70]
- S. Fiorucci et al.	- TNBS-induced colitis in BALB/c mice	- Rifaximin	- Decreased colonic IL-2, IL-12, IFN-α and TNF- β - Reduced colon MPO activity	[71]
*Probiotics and LBPs*				
- M. Bermudez-Brito et al.	- Wild-type Balb/c mice	- *L. plantarum* WCFS1	- Polarization of antigen-specific T cell response towards Th1, Th2, Th17 or Treg.	[73]
- S. M. Lim et al.	- LPS-stimulated murine peritoneal macrophages - Mice with carrageenan-induced hind-paw oedema - TNBS-induced colitis	- *Lactobacillus fermentum* IM12	- Downregulation of the NF-kB signalling, suppression of colon shortening, MPO activity as well as IL-6 and IL-17 levels - Inhibition of the expression of iNOS, COX-2, activation STAT3	[74]
- Q. Zhai et al.	- LPS-treated C57BL/6 mice	- *Lactobacillus salivarius*	- Reversed LPS-induced alterations in gut barrier function, colonic histopathology, Treg/Th17 balance, colonic immunomodulatory indicators	[75]
- S. E. Jang et al.	- TNBS-induced colitis in C57BL/6 mice	- *Bifidobacterium longum* LC67 - *Lactobacillus plantarum* LC27	- Inhibition of NF-κB pathway in macrophages and epithelial cells - Reduction of *Enterobacteriaceae*, particularly *Escherichia coli* and gut microbiota LPS levels - Increased lactobacilli and bifidobacteria - Inhibition of Th17 cell differentiation and RORγt expression - Enhanced Treg differentiation and Foxp3 expression - Restoration of the suppressed TJP expression - Increased IL-10 expression	[76]
- L. Zhou et al.	- DSS-induced colitis in BALB/c mice	- *Bifidobacterium infantis*	- Increased weight, decreased DAI and histological damage scores - Increased protein expression of Foxp3, PD-L1, as well as IL-10 and TGF-β1	[77]
- A. Steimle et al. - A. Rodríguez-Nogales et al.	- WT C57BL/6 and TLR5-deficient animals as well as BMCM, C57BL/6N (WT) and Tlr5−/− mice treated with DSS - DSS-treated C57BL/6J mice	- *Escherichia coli nissle* EcNΔfli) - *Escherichia coli Nissle* 1917	- Activation of TLR-5, resulting in IL-22-mediated protection against colitis - Reduction in clinical and histopathological signs of colitis - Preservation of intestinal permeability - Reduced levels of neutrophils, eosinophils, chemokines and cytokines (il-1β, il-12 mRNA levels) - Increased Treg cells - Restoration of tgf-β expression - Increased expression of MUC-2	[78,79]
- R. Mariman et al. - T. Mimura et al.	- Healthy BALB/c and C57BL/6 mice - IBD Patients	- VSL#3	- Downregulated Il13 and Eosinophil peroxidase - Upregulated Il12rb1 Cxcr5, Cxcr3, Cxcl10 in BALB/c mice - Increased B cells - Decreased T cells - Increase cluster of differentiation (CD) 11c(+) cells - Increased frequencies of Th17 and Treg in the MLNs - Maintained remission at one year in the 85% of patients - High IBDQ score	[80,81]
- Zhou, H. et al.	- DSS-induced colitis BALB/c mice	Saccharomyces boulardii	- Reduced body weight loss, DAI and histological score - Reduced EMT and decreased expression of VEGF	[82]
*Prebiotics*				
- K. Ishisono et al.	- TNBS- or DSS-induced colitis in C57BL/6 mice	- Pectin	- Decreased colonic IL-1β and IL-6 levels - Increased faecal concentration of propionic acid	[83]
- Grabinger T. et al.	- B6.129P2-Il10tm1Cgn/J mice	- 2-fucosyllactose	- Reduced histological scores, colon shortening - Decreased IL-1 β and IL-6 expression - Increased TGF-β and occludin expression - Expansion of the commensal *Ruminococcus gnavus*, accompanied by an enhanced caecal concentration of propionate	[84]
- S. Kanwal et al. - S. Kanwal et al.	- BALB/c mice treated with clindamycin and metronidazole - DSS-induced colitis BALB/c mice	- *Dictyophora indusiate*	- Reduced pathogenic bacteria (*Enterococcus*, *Bacteroides* and *Proteobacteria*) - Increasing beneficial bacteria (*Lactobacillaceae* and *Ruminococaceae*) - Decreased TNF-α, IL-6, and IL-1β levels - Increased expression of TJP (claudin-1, occludin, and zonula occludens-1) - Alleviated clinical, histological symptoms of colitis - Reduce MPO levels, NO activity - Elevated T-SOD levels - Reduced TNF-α, IFN-γ, IL-1β, IL-6, and IL-17	[85,86]
- C. Diling et al. - Y. Ren et al.	- TNBS-induced colitis Sprague–Dawley rats and Kunming mice - DSS-induced colitis C57BL/6 mice	- *Hericium erinaceus*	- Normalization of IL1α, IL-2, IL-8, IL-10, IL-11, IL-12, TNF-γ, TNF-α, VGEF, MIP-α, M-CSF and MPO levels - Increased Foxp3- and IL-10- positive cells - Reduced TNF-α and NF-κB p65- positive cells - Reduced proinflammatory microbes Enriched anti-inflammatory microbes - Downregulated NO, MDA, T-SOD and MPO - Reduced IL-6, IL-1β, TNF-α, COX-2 and iNOS	[87,88]
- R. Li, et al. - R. Li, et al.	- DSS-induced colitis C57BL/6J mice	- Muscadine grape (Vitis rotundifolia) or wine phytochemicals - Whole muscadine grapes (FMG) or dealcoholized muscadine wine (DMW)	- Decreased MPO activity, IL-1β, IL-6, and TNF-α levels in colon - Down-regulated NF-κB pathway by inhibiting the phosphorylation and degradation of IκB - Increased butyric acid and acetic acid faecal levels - Increased fecal IgA and mucin 2 - Decreased relative abundance of *Clostridium* and *Akkermansia* - Increased abundances of *Roseburia, Anaerotruncus*, and *Coprococcus*	[89,90]
*Synbiotics*				
- T. Shinde et al.	- DSS-induced colitis in C57BL/6J mice	- *B. coagulans* MTCC5856 + whole plant sugar cane fibre - *B. coagulans* + green banana resistant starch (GBRS)	- Ameliorated DAI and histological score - Preserved TJP expression - Restored serum IL-1β, IL-10, and C-reactive protein levels - Increase of the SCFA	[91,92]
- H. K. Altun, et al.	- UC patients with mild-to-moderate activity	- *E. faecium, L. plantarum, S. thermophilus, B. lactis, L. acidophilus, B. longum* + FOS	- Ameliorated disease status - Significant reduction of systemic inflammation (serum CRP levels)	[93]
*Postbiotics*				
- Tsilingiri K, et al.	- Organ culture system of human healthy and IBD intestinal mucosa	- *Lactobacillus paracasei* supernatant	- Reduced TNF-α production and most of the cytokines and chemokines involved in the pathology of IBD including CCL4, CCL2, IFNy and IL23p40.	[94]
- F. Yan et al. - X. Shen et al.	- Young adult mouse colon (YAMC) epithelial cells or kinase suppressor of Ras−1 knockout (KSRI−/−) mouse colon epithelial (MCE) cells - TNBS and DSS-induced colitis wild-type C57BL/6 mice	- *Lactobacillus rhamnosus GG*-derived protein p40	- Enhanced body weight gain prior to weaning - Promoted functional maturation of the intestine, including intestinal epithelial cell proliferation, differentiation, TJP formation and IgA production - Downregulated TN-α and IFN-γ levels - Reduced MPO activity, TNF, KC, and IL-6 mRNA levels - IncreasedCD4+Foxp3+CD25+ cells percentages in the lamina propria of the small intestine and the colon	[95,96,97,98]
- W. I. Izuddin, et al.	- Lambs	- *Lactobacillus plantarum RG14*	- Decreased leukocyte, lymphocyte, basophil, neutrophil and platelets count - Increased IL-6 mRNA - Decreased IL-1β, IL-10, TNF mRNA levels - Upregulated TJP-1, CLDN-1 and CLDN-4 mRNA levels	[99]
- Q. Ren et al.	- DSS-induced colitis C57BL6/J mice	- Conjugated linolenic acid (CLNA) isolated from *Lactobacillus plantarum* ZS2058)	- Inhibited weight loss, DAI and colon shortening - Alleviated histological damage, protected colonic mucous layer integrity - Upregulated the concentration of TJPs (ZO-1, occludin, E-cadherin1 and claudin-3) - Attenuated levels of proinflammatory cytokines (TNF-α, IL-1β, and IL-6) - Upregulated the expression of the colonic anti-inflammatory cytokine IL-10 and nuclear receptor PPARγ - Increased the activity of oxidative stress related enzymes (SOD, GSH and CAT) - Decreased MPO activity - Rebalanced intestinal microbial composition increasing the α-diversity (increased abundance of *Ruminococcus and Prevotella)*	[100]
- L. Van den Bossche et al.	- DSS-induced colitis C57Bl/6J mice	- Ursodeoxycholic acid (UDCA) and its taurine (TUDCA) or glycine conjugates (GUDCA)	- Reduced body weight loss, colonic shortening, and expression of inflammatory chemokines and cytokines, such as CXCL1, G-CSF, and IL-6 - Downregulated levels of CXCL1 and G-CSF - Increased *Firmicutes* to *Bacteroidetes* ratio - Increased abundance of *Akkermansia muciniphila*	[101]
- L. Zhou et al.	- TNBS-induced colitis C57BL/6J mice and Sprague-Dawley (SD) rats	- Butyrate produced by *F. prausnitzii*)	- Maintaining of Th17/Treg balance - Inhibited IL-6/signal transducer and the STAT3/IL-17 pathway - Promoted Foxp3 by targeting HDAC1	[102]
- R. Simeoli et al.	- DSS-induced colitis in BALB/c mice	- N-(1-carbamoyl-2-phenylethyl) butyramide (FBA)	- Decreased polymorphonuclear cell infiltration score - Reduced inducible NOS protein expression, CCL2 and IL-6 transcripts - Increased TGF-β and IL-10 levels - Limited neutrophil recruitment, recovered deficiency of the butyrate transporter, improved intestinal epithelial integrity and restored the distribution of occludin and ZO-1 - Inhibited histone deacetylase-9 and to re-establish H3 histone acetylation - Inhibited NF-κB and up-regulated of PPARγ	[103]
- Z. Zha et al.	- DSS-induced in C57BL/6 mice	- Xylan butyrate ester (XylB)	- Reversed the imbalance between IL-1β, TNF-α, IL-17A and IL-10 - Decreased relative abundance of *Oscillibacter, Ruminococcaceae* UCG-009*, Erysipelatoclostridium*, and *Defluviitaleaceae* UCG-01 - Increased butyrate content - Upregulated G-protein coupled receptor 109A protein expression - Inhibited HDAC activity - Activated autophagy pathway and inhibited NF-κB	[104]
- S. Facchin et al.	- IBD patients	- Microencapsulated form of sodium butyrate (MSB)	- Increased SCFA-producing bacteria *(Butyricicoccus and Subdoligranulum* in CD patients; *Lachnospiraceae* in UC patients) - Reduction of calprotectin levels	[105]
*Faecal Microbiota Transplantation*				
- C. Burrello et al.	- DSS-induced acute experimental colitis in C57BL/6 mice - DSS-induced chronic experimental colitis in C57BL/6 mice	- FMT by oral gavage of mucus and faeces from donor B6 mice	- Decreased levels of pro-inflammatory cytokine such as TNF, IL1β and IFNγ - Higher amounts of colonic IL-10 as well as increased frequencies of IL-10-producing APC and CD4+ T and iNKT cells - Reduction of F4/80+ macrophages, CD11b+Ly6G+ neutrophils and dendritic cells - Restoration of functional normobiosis, - Increased in colon length - Decreased expression levels of Ifng, Tnf, Il1b, Il17, and Il6, Camp, S100a8 and Muc1, Muc3, and Muc4 genes - Lower proportion of CD4+ T and CD8+ T cells expressing CD107a - Reduced numbers of colonic MHC-II-expressing professional APCs	[106,107]
- V. Jacob et al.	- UC patients’ mucosal biopsies	- FMT by colonoscopy	- Reduced Treg numbers - Reduced IFNγ - No differences in IL-4, IL-17, IL-22 or Th17	[108]
- STOP-Colitis study	- UC patients	- FMT by colonic route and nasogastric tube	- TBD	[109]
- FMTFUC study	- UC patients	- FMT by gastroscopy	- Decreased clinical index scores for diarrhea, abdominal pain, and blood stool - Normalized gut microbiota composition	[110]
- IMPACT-Crohn study	- CD patients	- FMT by colonoscopy	- Crohn’s Disease Endoscopic Index of Severity decreased - Increased alpha diversity - Higher colonization by donor microbiota was associated with maintenance of remission	[111]

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
