# Peer review of "The Role of Gut Microbiota Biomodulators on Mucosal Immunity and Intestinal Inflammation"

_cells, 2020, doi:10.3390/cells9051234_

Round 1
Reviewer 1 Report
This is an informative review, I really enjoy reading it.
The authors describe the mucosal immune dysfunction by focusing on the immune cells as the main cause of IBD.
This reviewer completely agrees with the authors' description, however, the epithelial cell function is also important for composing the mucosal immune system. Therefore, this reviewer recommends that the authors add a description regarding the role of epithelial cells in the mucosal immune system and the mechanism of breakdown of mucosal barrier function in IBD. This description makes a better understanding of the readers.
Next, in the section of postbiotics, they describe the mechanism of Bifidobacterium longum. Regarding this probiotic, interesting data has been already published as follows: (1) Upregulation of T-bet and tight junction molecules by Bifidobactrium longum improves colonic inflammation of ulcerative colitis.(Inflamm Bowel Dis. 2009 Nov;15(11):1617-8.) (2) Efficacy of probiotic treatment with Bifidobacterium longum 536 for induction of remission in active ulcerative colitis: A randomized, double-blinded, placebo-controlled multicenter trial. (Dig Endosc. 2016 Jan;28(1):67-74.)
This reviewer recommends adding these papers into references.
Author Response
Response to Reviewer 1
We warmly thank the reviewer for enjoying reading our manuscript.
- According to the reviewer’ suggestion we added a paragraph from line 68 to line 87 describing the role of intestinal barrier breakdown in IBD.
- We thank the reviewer for the suggestion and, accordingly we added the new references to the manuscript
Reviewer 2 Report
- “as well as the basis for therapeutic restoration of homeostatic immune function by manipulating the gut microbiota through existing microbe-targeted therapies, including antibiotics, prebiotics, probiotics, and faecal microbiota transplantation.”
Cite the effect on improving dysbiosis of adalimumab (Adalimumab Therapy Improves Intestinal Dysbiosis in Crohn's Disease. J Clin Med. 2019 Oct 9;8(10). pii: E1646. doi: 10.3390/jcm8101646)
- In your interesting chapter “Mucosal immune dysfunctions and dysbiotic microbiota in IBD” try to discuss the possible mechanisms that could explain new onset IBD in patients treated with anti-IL17 drugs (see, among others, "New Onset of Inflammatory Bowel Disease in Three Patients Undergoing IL-17A Inhibitor Secukinumab: A Case Series. Am J Gastroenterol. 2019 Jan;114(1):179-180. doi: 10.1038/s41395-018-0422-z.)
- “Therefore, it is desirable to develop personalized strategies to identify which patients should be treated with which drugs [47] by leveraging a more physiological approach, patient centred, with limited toxicity and high cost-effectiveness”
Discuss some paper trying to find biomarkers for prediction to response to therapy (for example:
Bertani L et al. Assessment of serum cytokines predicts clinical and endoscopic outcomes to vedolizumab in ulcerative colitis patients. Br J Clin Pharmacol. 2020 Feb 6. doi: 10.1111/bcp.14235.
On-Treatment Decrease of Serum Interleukin-6 as a Predictor of Clinical Response to Biologic Therapy in Patients with Inflammatory Bowel Diseases. J Clin Med. 2020 Mar 15;9(3). pii: E800. doi: 10.3390/jcm9030800.)
- Discuss that, according to Cochrane meta-analyses, the clinical role of probiotics in IBD is very limited
- Discuss the difficult to use prebiotic in CD patients, especially with a stenotic phenotype
- Discuss the lack of data for FMT in CD and that slight clinical benefit among placebo in UC
Author Response
Response to Reviewer 2
We thank the reviewer for the insightful papers that we added to the manuscript improving our manuscript
- Line 187-190. We added a sentence and the suggested reference on the role of adalimumab on the restoration of gut microbiota eubiosis
- According to the reviewer’s comment we now discuss from line 100 to 104 the potential role of anti-IL17 drugs (e.g. secukinumab) in the onset of new IBD cases.
- We thank the reviewer for the comment. We added a paragraph from line 176 to 187 describing some predictive biomarkers to response to therapy citing also the suggested references.
- According to the reviewer’s comment we now state in line 190 “However, the efficacy of probiotics in IBD remains uncertain, since different meta-analyses showed that there were no significant differences of remission, relapse, and complication rate between probiotics and placebo group, thus, more evidences from randomized clinical trials (RCTs) are required”
- According to the reviewer’s comment we now state in line 190 “Nevertheless, although prebiotics have an excellent safety profile, they have been associated with symptoms of dose-dependent abdominal pain, flatulence, bloating, and diarrhea; therefore, insoluble fibre intake is discouraged in the case of significant intestinal stenosis in IBD patients”
- We agree with the reviewer’s comment and therefore we now report the limited data on FMT in CD and the potential inconsistent results obtained in some studies on the role of FMT in UC.